# The Artificial Intelligence-Enhanced Echocardiographic Detection of Congenital Heart Defects in the Fetus: A Mini-Review

**DOI:** 10.3390/medicina61040561

**Published:** 2025-03-21

**Authors:** Khadiza Tun Suha, Hugh Lubenow, Stefania Soria-Zurita, Marcus Haw, Joseph Vettukattil, Jingfeng Jiang

**Affiliations:** 1Biomedical Engineering Department, Michigan Technological University, Houghton, MI 49931, USA; ksuha@mtu.edu (K.T.S.); halubeno@mtu.edu (H.L.); 2Betz Congenital Heart Center, Helen DeVos Children’s Hospital, Grand Rapids, MI 49503, USA; stefania.soriazurita@helendevoschildrens.org (S.S.-Z.); marcus.haw@helendevoschildrens.org (M.H.)

**Keywords:** congenital heart disease, deep learning, artificial intelligence, fetal echocardiography

## Abstract

Artificial intelligence (AI) is rapidly gaining attention in radiology and cardiology for accurately diagnosing structural heart disease. In this review paper, we first outline the technical background of AI and echocardiography and then present an array of clinical applications, including image quality control, cardiac function measurements, defect detection, and classifications. Collectively, we answer how integrating AI technologies and echocardiography can help improve the detection of congenital heart defects. Particularly, the superior sensitivity of AI-based congenital heart defect (CHD) detection in the fetus (>90%) allows it to be potentially translated into the clinical workflow as an effective screening tool in an obstetric setting. However, the current AI technologies still have many limitations, and more technological developments are required to enable these AI technologies to reach their full potential. Also, integrating diagnostic AI technologies into the clinical workflow should resolve ethical concerns. Otherwise, deploying diagnostic AI may not address low-resource populations’ healthcare access disadvantages. Instead, it will further exacerbate the access disparities. We envision that, through the combination of tele-echocardiography and AI, low-resource medical facilities may gain access to the effective detection of CHD at the prenatal stage.

## 1. Introduction

According to data from the Centers for Disease Control, congenital heart defects (CHDs), or diseases arising from abnormal heart formation, afflict approximately 1% of newborns annually in the United States [1,2]. CHD can involve a problem with the structure or function of the heart, altering normal blood flow. Some CHDs, such as single ventricle defects, represent critical heart conditions. Therefore, a swift prenatal diagnosis and timely interventions can reduce associated morbidity and mortality.

Echocardiography is a non-invasive diagnostic ultrasound imaging technique widely used to screen for structural and functional anomalies. Obstetric fetal CHD screening commonly includes a four-chamber (4ch) view and an outflow track evaluation. In contrast, a fetal echocardiogram is a dedicated and focused diagnostic ultrasound of the fetal heart performed and analyzed by subspecialized pediatric congenital cardiologists and sonographers. According to a meta-analysis by Zhang et al. [3], prenatal echocardiography for CHD demonstrates only moderate sensitivity (68.5%). However, the sensitivity for detecting CHD within low-risk pregnant women (i.e., in a typically obstetric setting) is considerably lower (36.1% [3]), suggesting that a significant portion of CHDs may be missed during the regular obstetric ultrasound [4,5,6].

Approximately one-sixth to one-fourth of these unidentified infants have a critical CHD, which is defined as a severe and life-threatening heart condition requiring an intervention in the first month of life [7]. Unfortunately, 25% of infants with an unidentified CHD are missed in postnatal screening and are discharged from the hospital [7,8], resulting in many infant deaths in the United States [2]. Missed critical CHDs are considered a gap in the current workflow of the diagnosis of CHD.

In the era of artificial intelligence (AI), AI-assisted technologies, including advanced algorithms, analytics, and imaging technologies, can potentially impact clinical practices in radiology. At the same time, these technologies hold promises for advancing fetal and neonatal care, particularly for closing the “diagnostic gap” mentioned above.

This article reviews recent developments and elevated capabilities in the use of AI with echocardiography for the prompt and precise prenatal detection of CHDs. The rest of this paper is structured as follows: Section 2 provides technical background on AI applications in echocardiography, followed by a discussion in Section 3 on how AI technologies are utilized to enhance the echocardiography-based diagnosis of CHDs. Section 4 presents a critical analysis of existing studies, followed by a discussion of ethical concerns in Section 5. Section 6 provides the current limitations of AI for detecting CHDs before the closing remarks in Section 7.

## 2. Technical Background

### 2.1. Techniques of AI

AI is a broad term covering machine-based algorithms and models designed to mimic human logic and intelligence. Machine Learning (ML) is a subset of AI [9]. In the last few decades, ML has increasingly been used in biomedicine to detect pathological conditions and assist in making diagnoses [10,11,12]. In this context, ML algorithms teach a computer to process echocardiographic data and extract useful information or features (also known as pattern recognition [13]), just like a medical doctor does, so the computer can predict the presence of a CHD, given the echocardiographic data.

ML can be divided into subfields that include (1) supervised learning, (2) unsupervised learning, (3) semi-supervised learning, and (4) reinforcement learning, as shown in Figure 1.

In supervised learning, a computer is trained to learn features or information associated with labeled data, i.e., echocardiograms with or without CHD diagnosed by medical doctors. Combining these learned features allows ML algorithms to make predictions or decisions. Logistic regression, random forests, gradient boosting, and support vector machines are commonly used supervised learning models. After an ML model has been trained, the model’s accuracy is tested with an unseen dataset [14,15]. Supervised learning inherently relies on feature extracts, so the training outcomes may be sensitive to input data.

The second ML method, unsupervised learning, allows a computer algorithm to discover correlations or relationships among unlabeled data [15]. Unsupervised ML algorithms can be applied to clustering, finding similarity, and dimensionality reduction. Because the true answers from data are unknown, quantitative performance assessments of these unsupervised ML methods are not often sought. However, such assessments become essential in biomedical research when labeled data becomes infeasible and when the assessments would offer insight into biomedical data. Investigating the plausibility and salience of identified correlations or relationships is still useful for knowledge discovery.

Semi-supervised learning falls between supervised and unsupervised learning. Semi-supervised learning uses labeled and unlabeled data to train ML models; it is often used when there is not enough labeled data, largely due to the costs associated with generating them. Consequently, semi-supervised learning can use a small amount of labeled data to guide model training, while a large amount of unlabeled data can refine the model.

The last subfield in ML is reinforcement learning (RL), often used in robotics and control systems. In this subcategory, the algorithm interacts with the environment, receives penalties for specific actions, and learns through trial and error. A non-technical description of RL can be found in Appendix A.

Deep learning (DL), a popular ML subfield, is often tied to neural networks mimicking how the human brain works. In DL, the term “deep” indicates the network’s depth or the number of layers of these “artificial” neurons. “Artificial” neurons can be activated, and connections among these neurons are associated with weights. DL models need to be trained with a lot of data, akin to teaching a child to identify an apple by showing them many pictures of apples. The training process allows a DL network to achieve a pre-determined goal. Because the training must be cast in a mathematical optimization framework, the pre-determined goal must be written out as a function known as a loss function.

During training, the DL model makes predictions based on the input data, and we measure its performance using the loss function. The lower the loss, the better our model performs. This process, also known as “backpropagation”, helps minimize the loss function by adjusting the DL model’s internal parameters (i.e., weights and neuron activation).

Loosely speaking, a trained DL model acts like a carefully designed filtration system; different neuron layers in the system represent varying features extracted from the input data. For example, suppose we input an echocardiogram into a DL model. In that case, some early layers might recognize simple features, such as the gross shape (e.g., heart chambers, valves, etc.), and later layers might learn how to combine separate features to resemble the object of interest, i.e., the heart and a diagnosis associated with the heart. DL models are widely used for medical applications, such as medical image processing, speech recognition, and biomedical signal processing.

It is worth noting that, as shown in Figure 1, DL is a methodology parallel to the four categories mentioned above. In other words, DL models can be trained under supervised, unsupervised, or semi-supervised frameworks. The training process iteratively adjusts weights associated with neurons. More recently, the DL methodology has been expanded into reinforced learning as well, as shown in Figure 1.

Notably, convolutional neural networks (CNNs) are a class of core DL methods for handling imaging data. A non-technical explanation is provided in Appendix A. CNNs have numerous applications in medical imaging, particularly in image classification, object detection, and image segmentation. Specifically, variants of CNNs have been used to calculate biometric features, segment lesions, detect anatomical structures, and control image quality in conjunction with fetal echocardiography, as reviewed below.

To this end, in AI-augmented echocardiography, one common approach is first to process ultrasound images or videos. Then, these preprocessed ultrasound data are trained and validated to build deployable AI models that can be translated into the clinical setting. The choice of the actual ML algorithms depends on the specific clinical application and clinical data, as reviewed below.

### 2.2. Echocardiogram Background

Echocardiography employs ultrasound technology to image cardiac structure and function, and it is an essential tool in diagnosing, assessing, and managing heart conditions. This technique utilizes soundwaves ranging from a 2 to 12 MHz frequency by an ultrasound transducer (also known as a probe), with higher frequencies enhancing the image resolution at the expense of decreasing the image depth and balancing clarity with anatomical coverage. During echocardiography, the ultrasound transducer transmits acoustic waves. The acoustic waves travel through the biological tissues and often reflect and scatter due to differences in acoustic impedance. The returned echoes (through reflections and scattering) can be received by the same ultrasound transducer and displayed as ultrasound brightness images to reveal tissue boundaries and internal compositions. These rendered brightness images of the heart are B-mode echocardiograms, enabling the real-time internal visualization of heart conditions and dynamics. Fetal echocardiograms are specialized applications of echocardiography meant to assess fetal heart structure and function during pregnancy (Figure 2).

These scans can be performed as early as 18 weeks of gestation, once heart structures have developed, thus allowing the early detection of CHD. If an issue is identified, ultrasound scans after 24 weeks are useful for initial treatment planning. During pregnancy, various imaging techniques can be used depending on the gestational age, including abdominal and endovaginal approaches, which involve differing the transducer placement on the patient. Sonographers view structures, such as the walls, septum, chambers, and valves, utilizing multiple views, such as the four-chamber view, abdominal situs view, left ventricular outflow tract view, right ventricular outflow tract view, three-vessel view, and trachea view, for comprehensive evaluation.

The sonographer manually operates frame selection, which is critical for cardiac image analysis. Different phases of the cardiac cycle are independently evaluated and selected to detect the end diastole frame, which denotes the largest cardiac cavity area, and the end-systolic frame, which is used to assess changes in cardiac dimensions and volume to evaluate heart function. The frame sets are used as reference points to determine cardiac biometrics. Each chamber and the wall separating them is individually assessed to detect any significant dropouts denoting septal defects. The chamber and valve movements are observed throughout the cardiac cycle to detect any size variations, conduction abnormalities, myocardium abnormalities, ventricular dysfunction, or abnormal position or the formation of the great vessels and valves.

## 3. Artificial Intelligence (AI) Applications

In recent years, numerous studies have been conducted on the effective use of AI methods in fetal echocardiography to diagnose congenital heart disease. We used relevant terms to search through WEB of Science, Scopus, and PUBMED using the following criteria: (1) publication data range from 2020 to the present (i.e., last five years); (2) the selected topic is related to applying AI to echocardiographic data for detecting CHD at the prenatal stage; and (3) the publication language is English. The details of key phrases used in the literature search can be found in Appendix B below.

The initial search was performed by two individuals (KTS and HL). They obtained 57 and 35 papers, respectively. Then, their lists merged, and they reviewed abstracts to determine their fitness for the abovementioned topic. After the abstract review, the number of papers was reduced to 27. Finally, three individuals (KTS, HL, and JJ) reviewed the twenty-seven papers to finalize their fitness to our topic, resulting in the twenty papers in Table 1 below.

Table 1 briefly outlines these twenty papers. Our review below focuses on the effectiveness and relevance of AI technologies in detecting CHDs.

### 3.1. Image Quality Control and View Selection

Image quality is crucial in detecting anatomic variations and diagnosing abnormalities in fetal echocardiographic images. Echocardiography quality is influenced by involuntary fetal movements, unusual heart size, ultrasonic speckle noise and artifacts, and the knowledge level of the fetal echocardiographer or primary physician [38]. An image quality control framework is essential for obtaining the best results in the fetal echocardiographic detection of CHDs. This research has motivated several recent studies.

Dong et al. developed a framework composed of three CNN-based networks (B-CNN [basic CNN], D-CNN [deeper CNN], and ARVB Net [aggregated residual visual block network]) for quality control in the four-chamber detection of the fetal heart. Their experiment on fetal echo datasets shows the framework’s effectiveness, especially ARVBNet, for image quality control. The algorithm provides a quantitative assessment and real-time image optimization information for novices. Although a three-step framework showed superior accuracy, an error in the first rough classification can affect the overall performance. The authors suggest that a multi-task learning model studied in future research may solve this problem to an extent [23].

Other researchers also focused on developing the quality of the four-chamber view. Notably, Qiao and colleagues creatively used generative adversarial network models to improve the quality of four-chamber view echocardiograms [16,24]. In the first study, Qiao et al. created a Pseudo-Siamese Feature Fusion Generative Adversarial Network (PSFFGAN) to synthesize four-chamber views with artifacts, ultrasonic characteristics, and speckle noises. Specifically, the PSFFGAN uses a Triplet Generative Adversarial Loss Function (TGALF), which optimizes the PSFFGAN to completely extract the cardiac anatomical structure information and obtain high-quality fetal four-chamber views using four-chamber sketch images [16]. In their second study, Qiao et al. created an AI model to collect diversified and high-quality fetal four-chamber views. The Enhanced-Style Consistent Modulation (ESCM) module learns two sets of modulation parameters for sketch and semantic contours. They also added a Generative Adversarial Network that performs image synthesis and quality evaluation at different resolutions. Their proposed sketch mass loss function effectively suppresses the loss values in the training process of the four-chamber detection [24].

Another application of AI in fetal echocardiography quality control is enhancing echocardiograms by increasing the brightness and contrast. Sutarno et al. built a CNN-based Low Light Image Enhancement (LLIE) architecture to enhance the light intensity of echocardiographic images. They also added a CNN-based classifier named “FetalNet” to create a stacked architecture that could improve the success rate of fetal heart defect prediction. The model (post-enhancement) could increase the success rate by approximately 25% and produce 100% sensitivity using testing data [19].

Selecting high-quality views from clinically acquired 2D echocardiograms is critically important to diagnose CHD accurately. The work by Li et al. [21] is promising. They proposed a DL neural network model that can simultaneously perform multiview classification and image quality assessment. Their deep learning neural network model consists of a backbone network, a neck network, a view classification branch, and a quality assessment branch. The backbone network is for feature extraction, while the neck network fuses features. The view classification branch can classify echocardiograms into seven different views: axial four-chamber (A4C), parasternal view of the pulmonary artery (PSPA), parasternal long axis (PLAX), parasternal short axis at the mitral valve level (PSAX-MV), parasternal short axis at the papillary muscle level (PSAX-PM), parasternal short axis at the apical level (PSAX-AP), and other views. Using the 170,311 echocardiograms acquired, their testing results showed an overall classification accuracy of 97.8%.

Wu et al. also applied DL technologies to recognize anatomical structures for view selections [34]. Specifically, they attempted to recognize five different views: apical four-chamber view, three-vessel view, three-vessel trachea view, right ventricular outflow tract view, and left ventricular outflow tract view. Overall, their method achieved remarkable performance, i.e., high sensitivity (75–100%), high specificity (>93%), and high positive predictive value (90.0–99.5%).

### 3.2. Detection and Classification of Fetal Heart Defects

The AI detection of fetal heart defects from echocardiograms is a vital step in classifying features for the potential diagnosis of CHD and other abnormalities. To accomplish this, several methods have been developed to obtain the possible views and features of fetal echocardiography with their inherent benefits and drawbacks. To build a DL model capable of image analysis, it needs to be trained using a controlled dataset of varying types. This would set the benchmark for the DL to complete the task when correctly asked. These datasets must be sufficiently large to configure the DL properly; however, there is a decrease in benefits for each addition. Several types of DL algorithms attempt to identify fetal hearts differently and are, therefore, trained uniquely.

Truong et al. [25] propose using ML with a random forest algorithm framework that employs tenfold cross-validation to train models for assessing the presence of CHDs. This method demonstrates that ML using RF algorithms can improve sensitivity with a value of 0.85, a specificity of 0.88, positive prediction rates of 0.55, and a negative prediction rate of 0.97 to detect CHD (B). Wong et al. [26] utilized a version of V-Net deep learning for multiview and multiclass image segmentation. Through the careful selection of datasets based on images following verified landmark conditions, they showed how V-Net alone can accurately segment fetal echocardiograms in multiple views and abnormal conditions. Their training of the modified V-Net used 598 images with 14 anatomical labels, excluding some abnormalities that did not contain the structure necessary for the label, between the gestational ages of 18 + 0 and 24 + 6 weeks. These techniques have varying benefits, especially with image analysis and potential differences in the resource draw and needs. Various detection methods can be more or less suited for the classification of different fetal heart features from different views. The classification of different features, such as septum, heart walls, ventricles, and vessels, from different views is vital for diagnosing CHD. They provide the information necessary to derive a diagnosis of CHD or other potential abnormalities [28]. Wong et al. used a method that searches for 14 anatomical landmarks in a combined view model of 4CHV and 3VTV. They noted that some segmentations are more difficult to segment out, such as the trachea, interatrial septum, and mitral valve, with Dice coefficients lower than 70%. They successfully segmented anatomical landmarks within a 1% Dice coefficient of both separate models’ available landmarks, concluding that using multiple models for multiple views is unnecessary.

### 3.3. Measurements of Cardiac Function and Parameters

Defining the heart’s condition through the measurement and assessment of fetal biometry allows for reliable diagnostic procedures to occur. The reliable positive and negative prediction of CHD and other abnormalities in the fetal heart enables crucial early intervention and treatment.

Several different measurements are used to quantify the possibility of CHD or other abnormalities. Doppler traces can show the flow from the aortic arch and arterial duct, across the oval foramen and in the pulmonary veins, inflow through the atrioventricular valves, and outflow through the arterial valves. The verification and measurement of cardiac structures and volumes are crucial for diagnosis. The identification of anatomical landmarks and the quantization of their size are achieved through methods such as cropping and segmentation [28]. L. Yan et al. used a two-stage method to mark and segment three vessel samples through an object identification tool called Yolov5, which localizes the Region of Interest (ROI) within the original full-sized echocardiographic images. Subsequently, a modified Deeplabv3 equipped with their novel AMFF (Attentional Multi-scale Feature Fusion) module is applied in the second stage to segment the three vessels within the cropped ROI images. Furthermore, measurements can be drawn from time intervals in echocardiography and are used in several measurements to observe periods of activity within the fetal heart, as many actions within the fetal heart are periodic.

The segmentation of a view identifies anatomical landmarks, from which measurements can be extrapolated. As shown in Figure 3, the segmentation of the heart from an echocardiogram can be achieved by either a DL segmentation algorithm (e.g., ZoomNet [39]) or a DL object detection algorithm (e.g., YOLO [40]).

Measuring the parameters and functions identified and quantified in the AI detection of fetal biometry is integral to diagnosing positive and negative predictions of CHD. Many of these measurements are made to encompass the possibility of abnormalities present or absent in a cardiac system. Without a comprehensive assessment through various parameters, CHD detection is less reliable.

### 3.4. Diagnosis of Specific Abnormalities

The analysis of a specific abnormality in the fetal cardiac structure is significant in determining congenital heart disease. AI can assist physicians in diagnosing malformations in the fetal heart. Ventricular septal defect (VSD) is a deformity in which there is a deficiency in the wall separating the right and left ventricles. Small- to medium-sized VSDs are missed even by expert fetal cardiologists. Atrioventricular septal defects (AVSDs) are more serious defects that, if left undetected, can lead to severe consequences after birth. Several studies have focused on applying AI models in fetal echocardiographic images to detect these defects. Veronese et al. analyzed the performance of Fetal Intelligent Navigation Echocardiography (FINE) used for the prenatal identification of AVSDs. They used spatiotemporal image correlation (STIC) volume datasets from four-chamber views. This navigation system successfully determined the nine standard fetal views after it was applied to the input datasets. The accuracy was 100%, as it was able to identify abnormal fetal hearts with atrioventricular septal defects and a common atrioventricular valve [36].

Tetralogy of Fallot (TOF) is also a severe condition that can lead to infant death resulting from pulmonary artery obstruction, right ventricle thickening, septal anomalies with an overriding aorta, and many other anomalies. In severe cases, TOF has higher mortality rates than VSD. Yu et al. differentiated TOF and VSD cases using four convolutional neural network models, VGG19, ResNet50, NTS-Net, and the weakly supervised data augmentation network (WSDAN), on fetal echocardiography images. WSDAN, with an AUC of 0.873, showed the best performance among all the models in classifying the images [18].

Another kind of CHD found in infants is total anomalous pulmonary venous connection (TAPVC). Wang et al. aimed to predict TAPVC using a parameter called the PLAS ratio from fetal echocardiography. They analyzed if the calculation of the PLAS (the ratio of the epicardium-descending aortic distance to the center of the heart-descending aortic distance) using CNN would be a useful feature to detect TAPVC. The dataset included echocardiographic images of 287 fetuses with normal hearts and 32 fetuses with isolated TAPVC (25.6 ± 2.7 weeks). The computerized calculation used five DL segmentation models, including DeepLabv3+, FastFCN, PSPNet, and DenseASPP. The results showed that the PLAS ratio was higher in the TAPVC cases than in the controlled group in both manual and DL processes [17].

One of the most critical CHDs is hypoplastic left heart syndrome (HLHS). Day et al. trained CNN to automatically detect fetal HLHS using a specific single-center fetal database. This retrospective fetal database included 161 fetal patients and provided single-expert-identified ‘ideal’ four-chamber (4ch) view frames to identify and extract up to 10 frames chronologically before and after the ‘ideal’ 4ch frame. A total of 10,248 frames were obtained and analyzed (3960 with HLHS). Their results suggest that while specific AI models are not ready or good enough to operate independently, they will augment human performance [31].

In another study by Li et al. [33], AI detected VSDs very accurately, i.e., mAP@0.5 (a variant definition of accuracy) reaching 0.926. Interestingly, under AI guidance, the VSD detection accuracy improved by 6.7% and 2.8% among junior- and intermediate-level doctors, respectively.

### 3.5. Identification of Normal Heart from CHD

Accurately distinguishing normal fetal hearts from complex congenital CHDs is an essential and widespread need. Poor sensitivity in this task can reduce treatment options, negatively impact outcomes, and lead to unnecessary tests and referrals. Several studies have created AI models that differentiate between normal and abnormal fetal hearts. Arnaout et al. [20] showed that trained ensemble learning models performed well in detecting CHD. They used CNN on multimodal images to segment different views of the fetal heart and then distinguish between normal hearts and complex CHD cases. The model achieved an AUC of 0.99, 95% sensitivity, and 100% negative predictive value, working on 4108 fetal test cases in classifying normal and abnormal hearts.

Using a variant CNN model, the You Only Look Once (YOLO) model [40] shows a noteworthy performance in identifying CHD. Yang at el. utilized the YOLOv5 model to distinguish normal and abnormal fetal heart ultrasound images accurately. The dataset included normal fetal cases and VSD test cases where YOLOv5 showed 92.79% accuracy in identifying abnormal fetal hearts [32]. In addition to a high accuracy and prediction rate, less complexity in the computational process is also a desirable factor in working with large datasets. Magesh et al. proposed a new model integrating deep Reg net with CNN to extract features from ultrasound images and classify them into normal and CHD with less computational complexity. The ultrasound images from 363 pregnant women were also preprocessed using a SCRAB filter to eliminate the noise artifacts. The model could classify the images into two groups with 98.4% accuracy in identifying the CHD cases. It showed better performance compared with other DL networks used in fetal heart CHD classification [35].

While DL models show a higher accuracy in classifying normal and abnormal hearts from echocardiographic images, considering high-volume clinical indicators can also be important for validating the results. Qu et al. performed a data analysis using ML on pregnant women who had fetuses with CHDs. An explainable boosting machine (EBM) was used for prediction, and around 1127 predictors were included, with the top predictors selected according to contributions and performances. The results showed 0.65 accuracy, 0.74 sensitivity, and 0.65 specificity, where maternal UA, glucose, and coagulation were the most consistent predictors [37]. In their study, echocardiographic data read by two experts were used as the ground truth for diagnosing CHDs.

Fetal arrhythmia may cause fetal death, abortion, or hydrops. Although electrocardiogram (ECG) is the leading modality for assessing heart rhythm it is technically challenging in fetuses. An echocardiogram can detect arrhythmias using M-mode across the atria and ventricle to detect an AV block or atrial arrhythmias. Pulse Wave Doppler (PWD), as developed by Yang et al., uses an intelligent quantification system (HR-IQS). Specifically, they used a DL-based method to detect PWD signals automatically. The arrivals of E, A, and V waves are detected, and fetal cardiac time intervals (CTI) are computed. The resultant HR-IQS system was validated using multicenter PWD data, specifically from 6498 PWD spectrums over the junction between the left ventricular inflow and outflow tracts from 2630 fetuses. They reported a “significant positive correlation (*p* < 0.001) and moderate-to-excellent consistency (*p* < 0.001) between the manual and HR-IQS automated measurements of CTIs” [22]. The preliminary results indicated that the trained HR-IQS system has the potential to detect fetal arrhythmia [22].

## 4. Critical Analysis

A detailed analysis of the eighteen (18) AI models reviewed above was conducted, and their characteristics are summarized in Table 2 below. Two papers in Table 1 are not included, as justified below. In [36], AI was applied to medical testing data, and echocardiographic data were used to verify the presence of CHD. Ref. [37] was a case study and only included four cases. Thus, neither study fits the scope of this critical analysis, in which echocardiographic data are used as inputs for AI.

### 4.1. Categories of AI Models

According to the American Society of Echocardiography (ASE), AI products can be divided into (1) Assisted, (2) Augmentative, and (3) Autonomous models [41]. In the first category, AI products will produce only clinically relevant data. For instance, an AI product measuring the deformation of the heart is considered an Assisted AI product. An Augmentative AI product analyzes raw echo data and produces clinically relevant data. Thus, if the heart deformation is converted to a summary metric reflecting myocardial infarction, the AI model will be elevated to an Augmentative model. Models in the first and second categories require physicians’ involvement in decision-making and reporting. In contrast, autonomous models will make clinically relevant decisions using an AI model without the involvement of a medical doctor.

Interestingly, there are no Augmentative AI models. Two AI models are synthetic echocardiogram generators that can be used for training. The rest of the papers are split between Assisted (7/14) and Autonomous (9/14) AI models.

### 4.2. Selection of Training Data

As shown in Table 2, only four studies (4/18) used more than 10,000 images for training. Among them, three studies included data from more than one ultrasound vendor. Nearly half of these studies (7/18) used data from a single center with small data sizes (<1000 echocardiograms). Using a large amount of data acquired across imaging vendors and multiple medical centers will enhance the validity of these ML algorithms.

Only a small fraction of these studies (approximately 10%; 2/18) provided sufficient information regarding data acquisition and collection methods. Evaluating ML models in CHD detection without knowing detailed information about the data hampers our efforts to truly validate these ML methods and translate them into the clinical workflow.

Unlike other ML-based medical research (e.g., segmentation of brain cancer), none of these studies used a public database. Although important, the establishment of public echocardiography databases for fetal applications is still in its infancy. Public databases with comprehensive data are crucial for further developing AI algorithms to detect CHD and understand the potential of developed ML models.

### 4.3. ML Models and Completeness of Their Descriptions

As shown in Table 1, all studies except one used DL-based methods. In particular, due to the tremendous achievements of CNN-based models in Radiology, most studies adopted one or more variants of CNN models. One of the significant advantages of CNN models is that features extracted from echocardiograms are automated. Thus, these models can be integrated into the clinical workflow with full automation. In contrast, the study by Truong et al. [25] adopted Random Forest, a non-DL-based method, to classify fetuses into normal and abnormal groups. Twenty (20) manually crafted features (e.g., cardiothoracic ratio, maximum Doppler Velocity at aortic artery, etc.) are fed into the Random Forest model. These manually crafted features prevent the proposed ML model from being fully automated unless fully automated methods are developed to create these features.

After reviewing the eighteen papers listed in Table 2, none of these studies performed a formal computational complexity analysis. We found that ML models can be divided into three groups based on the completeness of the description of the ML model. Source codes are provided by three (3 out of 18) papers, and readers can potentially verify their neural network configurations. In contrast, in four (4 out of 18) papers, three studies used neural network models, and their configurations were not discussed. One paper used a Random Forest classifier. Only two of the remaining eleven papers in the last group provided sufficient details to reproduce their neural network models. Although most of the other nine papers used published models, the information provided by the authors may not allow others to grasp their ML models fully. For instance, the classical ResNet model [42] has many variants. Simply stating that the ResNet model was used does not provide sufficient information for further assessments.

### 4.4. Generalization and Computational Requirements of Reviewed ML Models

Generalization, defined as AI’s ability to apply knowledge to new data [43], has been a key challenge in many diagnostic AI applications. Although no formal studies of AI generalization for detecting CHDs have been identified, most studies (14/18) only used data from one center, and only one-third of the studies used data from more than one (echocardiography) vendor. Also, two-thirds of the studies did not describe their data acquisition protocols. Alarmingly, one study [35] included little information about the data procurement process. Limitations in data collection pose difficulties in generalizing these AI models. Furthermore, as shown in Table 2, eight studies (out of 18) used a small amount of data (<1000 images). Thus, these models may be too closely tuned for the training data, i.e., overfitting, making the generalization of these models challenging. Consequently, we submit that more work is needed to demonstrate that the results of these studies (Table 2) can be generalized.

To improve the generalization, AI algorithms should be validated using diverse sources of data. Since collecting data and subsequently making accurate annotations requires considerable resources, data augmentation has been attempted. Sfakianakis et al. [43] used DL-generated synthetic ultrasound data to improve training for image segmentation. In contrast, Chen et al. [44] adopted more traditional image data augmentation methods, such as rotation, to improve the training of DL-based object detection. The GAN-based echocardiogram synthesis methods by others [16,24] can also be used for data augmentation. In the literature, advanced modeling methods [45,46] have been used to generate synthetic data for ultrasound elastography applications [47]. Given appropriate developments, these advanced modeling methods can also potentially train DL-based CHD detection using echocardiographic data.

Although data augmentation may be appropriate for certain applications, such as image segmentation and object detection (e.g., certain views of echocardiograms), the generation of pathologies requires more consensus. That is why federated learning [48] has been explored. Federated learning (also known as collaborative learning) is a specific setting where multiple institutions collaboratively train an AI model without requiring physical data exchanges and centralized data storage. Without the legal paperwork or procedures required to exchange data, the collaborative training of novel AI methods can be accelerated. Goto et al. [49] adopted federated learning to access data from Massachusetts General Hospital, the University of California San Francisco, and Keio University Hospital. Their results showed that access to data from two additional institutions improved the AI’s performance. Consequently, we think the federated learning approach is a practical approach by which computational scientists can access diverse data needed to generalize their AI models.

As we found in Section 4.4, many models did not provide sufficient information to assess their computational complexity formally. However, two-dimensional echocardiograms were mostly used as inputs for these AI models in all applications. Thus, these studies did not require high-end Graphic Processing Units (GPUs) (Table 2). Fifteen studies used non-professional-grade GPUs (i.e., gaming GPUs: GTX 1080, GTX 2080, RTX 3050, and RTX 3090; Nivida Inc., Santa Clara, CA, USA). Two studies did not disclose the GPU equipment. The remaining two studies used the V100 GPU (Nivida Inc., CA, USA), which only has an onboard GPU-accessible 16 GB memory. In 16 studies where the GPU details are provided, the GPU memory varies from 11 GB to 48 GB. This finding suggests that most AI models can be implemented into clinical workstations.

### 4.5. AI Performance Versus Clinician’s Performance in Clinical Workflow

In the eighteen studies shown in Table 1 and Table 2, no direct comparisons to clinical studies can be found for nine studies. The rest of the nine studies can be divided into three groups: (1) heart structure and vessel delineation [26,28], (2) anatomical measurement [17], and (3) CHD differentiation [20,25,31,32,33,35].

The first two studies [26,28] investigated the application of DL methods to segment either heart chambers or vessels connecting to the heart. Their Dice scores ranged from 0.74 to 0.89. The Dice score measures the (area) overlap rate between the AI-segmented structures/vessels and the ground truth (i.e., clinical experts delineated structure/vessel). A Dice score of one indicates a perfect overlap (i.e., identical), while a Dice score of zero denotes no overlap. The Dice scores ranging from 0.74 to 0.89 suggest that DL methods obtain highly consistent results compared to those obtained by clinical experts. Compared to the anatomical delineation using other modalities (i.e., often above 0.9), such as vessel segmentation using X-ray digital subtraction angiography [50], CT angiography [51], and intravascular ultrasound [52], and brain cancer delineation using MRI data [53], the Dice scores reported in [26,28] were slightly lower.

In one study by Wang et al. [17], in the second group, the post-left atrium space (PLAS) ratios detected by the AI method were comparable to those obtained by human experts (accuracy: 89% vs. 88% and area under the curve [AUC]: 0.88 vs. 0.87).

In the last group, six studies [20,25,31,32,33,35] investigated the detection of CHDs. Two of them were designed to detect specific CHDs. In the study by Day et al. [31] the sensitivity and specificity of the AI-based detection of HLHS were comparable to those reported in the clinical literature. For example, in the UK’s NHS fetal anomaly screening program, the sensitivity for detecting HLHS is around 92.7% [54], while the AI algorithm described by Day et al. has a sensitivity of 94.3%. Thus, the AI algorithm slightly outperforms the clinicians regarding the sensitivity for screening for HLHS. In another study by Li et al. [33], the detection of VSDs was investigated; since its performance metrics differed from the standard metrics in clinical studies, direct comparisons are difficult. However, its performance was excellent (e.g., a variant of the accuracy measure is close to 0.92).

The four studies [20,25,32,35] in the last group were designed to differentiate abnormal fetal hearts from normal ones. All these four studies performed well. Three studies [20,25,35] reported excellent AUC values (i.e., >0.94) and overall accuracy (i.e., >0.88), as shown in Table 3 below. In the other study by Yang et al. [32], the AUC was not reported and the total accuracy was 80%. The AUC values of these three studies are on par with the overall summary AUC value in a meta-analysis of echocardiography-based CHD detection studies [3]. Notably, Arnaout et al. [20] reported an AUC of 0.99, 95% sensitivity, 96% specificity (95% CI, 95–97%), and 100% negative predictive value in distinguishing normal from abnormal hearts.

The specificity values of these three studies were lower than the average specificity of the meta-analysis (93% [average of three studies] vs. 99.8%), as shown in Table 3. However, the sensitivity of the four AI studies was considerably higher (90.5% [average of four studies] vs. 68.5%). It is generally recognized that echocardiography’s sensitivity in screening CHDs varied significantly from prior studies (e.g., from 92% [55] to 16% [56], on average 68.5%). A lower sensitivity (on average 36.1%) typically comes from low-risk pregnant women (i.e., a typically obstetric setting), as the meta-analysis results suggested ([3]). Consequently, an AI-based detection method with a reasonable specificity value (e.g., 90%) can be developed into an effective screening tool in the obstetric setting.

### 4.6. One Case Study [57]

We found one study in which AI’s performance was compared with multiple human readers in a point-of-care setting for a low-risk population. Between 2015 and 2016, 108 fetuses (42 normal and 66 abnormal) from the northwestern region of the Netherlands were studied. CHDs were confirmed at birth. To mimic an initial screening of CHDs at the prenatal stage, stored images were shown to three fetal cardiac experts as live imaging in a point-of-care setting. The CHD status was hidden from all three fetal cardiac experts. It is also important to note that a DL model previously trained by Arnaout et al. [20] was used in this study. “all CHD status” was hidden from the DL model as well”.

During the initial clinical assessment by OB doctors, 31 abnormal cases (out of 66) were missed, translating to approximately a 50% miss rate. In this low-risk screen setting, a sensitivity of 50% is consistent with the results shown in Table 3. In contrast, the DL model’s sensitivity and specificity were 91% and 78%, respectively. Three blinded human experts achieved a sensitivity and specificity of 55 ± 10% (mean ± SD; range, 47–67%) and 71 ± 13% (range, 57–83%), respectively. They concluded that the previously trained DL model had higher sensitivity than the initial clinical assessment in detecting CHDs in the cohort of 108 fetuses in a low-risk population. Also, the DL model outperformed three fetal cardiac experts, particularly in sensitivity.

Moreover, these abnormal cases included 19 lesions the DL model had not encountered during its training. The encouraging outcome is that the DL model correctly identified 16 of them (out of 19), suggesting the training is sufficient. Recall that the DL model by Arnaout et al. [20] is one of the best DL models to detect CHDs (see Table 3).

### 4.7. Scientist–Physician Partnership

For all studies listed in Table 2, based on the authors’ affiliations, we found that 17 out of 18 publications resulted from collaborations between computational scientists/engineers and clinicians. This trend reflects the fact that applying AI models to CHD detection is multidisciplinary in nature; thus, collaborations between computational scientists and clinicians are needed to advance this field and translate AI technologies into the clinical workflow.

## 5. Ethical Concerns of Artificial Intelligence

AI has great potential to improve the detection of fetal CHDs; however, it also raises many ethical dilemmas. The first question that can be raised is the “black box” phenomenon. ML techniques, including DL in medicine, are heavily criticized for their “black box” nature since the way in which many ML models arrive at decisions cannot be explained, and the decision process is not transparent [58,59]. Transparent ML models ensure safety, fairness, and accountability, making them more attractive for medical doctors to use with confidence. This issue can only be solved using more interpretable models and transparency in the validation process. However, regarding detecting CHDs, the “black box” nature of ML models may not be as significant if we use an ML model as a screening tool. We can adjust the ML model to reach a high sensitivity (e.g., >99%) if the false positive rate is not too high (e.g., 30%).

Another ethical concern is that biases exist among ML models. As explained before, supervised ML models learn from input data. Thus, one primary source of biases stems from the biases presented in their training data, which may be related to race, ethnicity, genetics, region, or gender [60]. Moreover, ML models that use echocardiographic data may also need to sample training data across a spectrum of ultrasound vendors, as each vendor often uses proprietary techniques to process echocardiograms before they are displayed on its scanners. Furthermore, data from low-resource medical facilities are rarely used in all the studies evaluated; this is another form of data selection bias. To address the bias problem, we should consider a standard protocol for training data selection [60]. Creating open-source benchmark training data in which different CHDs from various patient populations are appropriately sampled may allow us to mitigate this bias problem. However, eliminating the sources of biases is challenging.

In the era of ML, deploying or training ML models relies on a huge amount of patient data. Consequently, ensuring patient privacy and confidentiality is a primary ethical concern. For instance, CHD detection may be enhanced if electrical health records (EHRs) can be added to the ML model. Then, should patients’ consent be required before using EHR data for building an ML model? The Health Insurance Portability and Accountability Act (HIPAA) requires such consent in the United States. However, such a requirement may not be universal. Patients should be allowed to opt out in countries where such consent is not required.

As shown in Table 2, many studies aimed to develop Autonomous AI models. These models may fundamentally change the patient–clinician relationship. Relying on Autonomous AI can misguide physicians, increasing the risk of causing harm and exposing them to potential legal action [61].

Additionally, there is the issue of fair accessibility [62]. For example, like many other modern technologies, the deployment of diagnostic AI typically starts with areas with sufficient resources. Thus, access to advanced diagnostic AI may further widen the healthcare disparities.

## 6. Discussion

### 6.1. General Challenges

Reviewing the available literature, we found that AI algorithms may aid in detecting and classifying CHDs. Although applying ML and DL algorithms in fetal echocardiography has numerous benefits, researchers have encountered several challenges.

First, as shown in Table 2, most AI models are not fully tested. Most studies were conducted with limited data (e.g., a single center and a small number of echocardiograms). The demographic information of these data is not clear. Moreover, biases in the training process cannot be evaluated without knowing the data selection. For instance, Taksøe-Vester et al. highlight the inevitable bias associated with supervised learning models, as these models are trained on datasets manually segmented by humans [63]. Thus, the superior accuracy reported by some of these studies may not be generalized. The development of such AI models will benefit from creating public databases with comprehensive patient information, allowing all AI models to be tested transparently.

Second, we observed that all the studies shown in Table 2 were outcomes of multidisciplinary collaborations. These collaborations should be applauded. Medical domain knowledge will assist computational scientists in developing better AI models. Also, the collective knowledge from physicians may help avoid the biases of the AI model and facilitate appropriate clinical decisions [58]. Along with the research on AI models used in fetal echocardiography, a framework for rigorous clinical assessment is also required [64]. Recall that the current testing metrics, such as sensitivity, accuracy, etc., are mainly designed for generic ML studies. Thus, metrics more suited for CHD detection can be further investigated.

Third, the adoption of AI-based CHD detection is still low. One limiting factor is the opaqueness of many AI models. The interpretation of AI algorithms is complex and operator-dependent, as “black box” models are created using raw data directly [14,15]. To overcome the “black box” issue and increase clinicians’ acceptance, steps should be taken to build clinically interpretable systems. For example, Moorfield’s Eye Hospital has developed a different approach. In its system, a segmentation network generates an intermediate tissue representation, allowing clinicians to visualize and quantify key areas of retinal pathology [65], increasing physicians’ confidence.

Our current study is a mini-review. Although we extensively searched the relevant literature, our selection could be biased, and our observations and opinions are bound by the papers selected.

### 6.2. Clinical Translation/Implementation

According to what was reported by Cardiovascular Business on 10 January 2025 [66], the US Food and Drug Agency (FDA) has cleared 161 AI products in cardiology up to the date of their publication. Our search found that one product, namely FETOLY-HEART, is relevant. FETOLY-HEART is an AI-assisted product based on ASE categorization [41] and can automatically detect certain heart views. Since this AI tool only received its clearance in August 2024, its clinical outcomes have not been reported in the literature.

Since most AI products for detecting CHDs have minimal or moderate risks, they are likely Class II medical devices and will only require FDA 510K clearance for regulatory approval. In other words, clinical trials may not be required to allow AI products for CHD detection to enter the market. That explains why very few clinical trials are known. Through a search of clinicaltrials.gov, we only found one clinical trial register, and the results have not been updated. Thus, we expect that more AI products of this kind will be cleared for clinical use in the near future.

Given the minimal or moderate risks involving in using AI products for detecting CHDs, their integration into the clinical workflow mainly resides in logistical issues, such as the integration into existing software platforms, e.g., picture archiving and communication system (PACS). Blezek et al. discussed various scenarios for deploying diagnostic AI products into the clinical workflow [67]. One scenario suggested by them may offer a plausible pathway for deploying an AI product aiming at CHD detection as summarized below.

If a cardiologist wishes to view the AI results, the cardiologist can launch an AI product by clicking a (software) button integrated into the PACS system. Then, the AI product opens and can automatically select one or multiple views to make a classification. Finally, the AI will show its results with some explanations. Cardiologists can see these AI results under a rapid review mode. The rapid review mode will include all selected images and the AI’s explanations during the decision-making process. Finally, the cardiologist can decide to accept or reject the AI’s results. Explained AI has been attempted by others [37,68]; these results are encouraging. Thus, its integration is possible and will enhance the usability of AI-based software for CHD detection.

## 7. Future Outlook and Closing Remarks

We consider AI-based CHD detection a leap forward for reducing diagnostic errors and improving clinical outcomes, particularly regarding sensitivity (see Section 4.5). If the work reviewed above is implemented into the clinical workflow, many studies will often require physicians to conduct many tasks manually (e.g., entering ultrasound measurements and selecting ultrasound images). Further developments can be focused on automation to streamline the workflow, including view selection, image enhancement, and quality assessment. Based on the studies reviewed, AI technologies for view selection, image enhancement, and quality assessment are available; thus, integrating these sub-components can offer a fully automated system for detecting CHDs in real time.

In addition to being used as a screening tool, another practical initial approach may be to set up AI platforms to identify specific CHDs that needs interventions soon after birth in a tertiary care setting (single ventricle, critical valvar stenosis, etc.). This would significantly enhance perinatal care and survival with CHDs.

Moreover, the availability of automated AI platforms for detecting CHDs could significantly benefit low-resource medical facilities, such as in rural areas and developing countries. As discussed above, although deploying diagnostic AI in low-resource areas, like ones for CHD detection, should be given priority, such deployment could be limited for financial reasons. Hence, the integration of AI into the traditional framework of tele-echocardiography [69] might bridge the gap for low-resource areas. In tele-echocardiography, an ultrasound technologist or radiologist first obtains cardiovascular ultrasound images from a patient. Then, these images are transmitted to a remote tertiary care setting for diagnosis, review, and image interpretation by a cardiologist. In this sense, an AI system can be deployed in a tertiary care setting and as a remote “cardiologist”, enabling rapid and accurate screening or decision-making. Indeed, the telemedicine [70] that has benefited low-resource areas may be further enhanced by the rapid development of diagnostic AI.

We conclude that AI-based CHD detection is feasible with the existing technologies. However, limitations restrict AI models’ potential for clinical applications and possibly dampen their clinical effectiveness. If these limitations are acknowledged and the process is approached cautiously, applying AI algorithms to detect and classify CHDs will be integral to future pediatric cardiac care.

Although the AI studies reviewed above have achieved encouraging results, there is still room for improvement. The improvement could be achieved if physiological data from the fetus and other medical test results were added to the AI-based predictive modeling. Recall that in [37], several medical testing results (e.g., maternal serum uric acid, glucose, etc.) were shown to be independent predictors of CHD. However, this is a single study, and others should verify their results. Also, the complex roles and associations of these biomarkers regarding CHD [71] should be further investigated. Furthermore, wearable biosensors can continuously collect physiological data about a fetus in the womb. In a recent publication, the American Heart Association’s Scientific Advisory Committee advocated using wearable technology to detect early signs of CHD [72]. Although wearable devices are routinely used to monitor maternal health, they have not been systematically used for detecting CHDs at the prenatal stage [73].

## Figures and Tables

**Figure 1 medicina-61-00561-f001:**
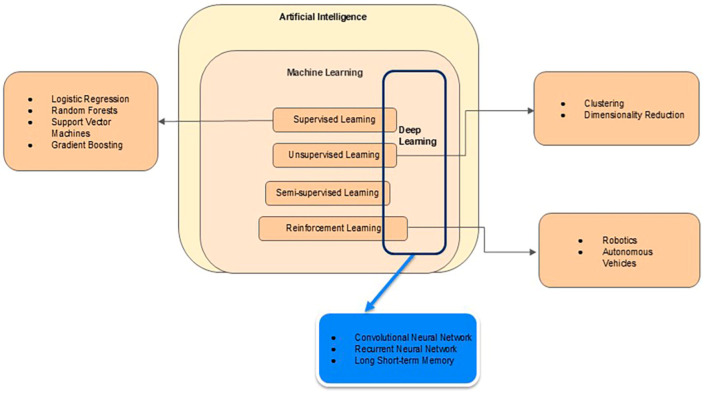
A schematic diagram showing the categorization of AI methods. Some AI models have been adopted and refined for their use in the detection of CHDs in conjunction with fetal echocardiography.

**Figure 2 medicina-61-00561-f002:**
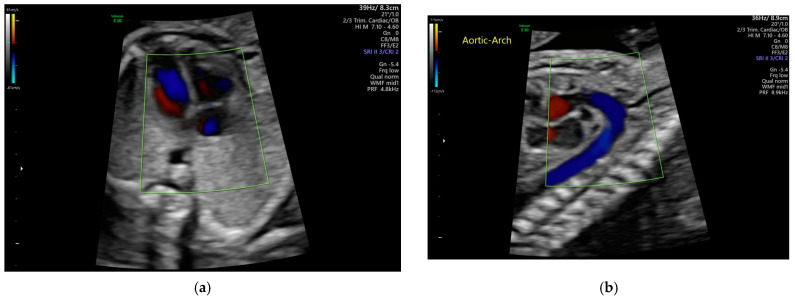
(**a**) An ultrasound image of a fetal heart in the four-chamber view and (**b**) an ultrasound image of a fetal heart in the aortic arch view. Both ultrasound images are overlaid with Color Doppler.

**Figure 3 medicina-61-00561-f003:**
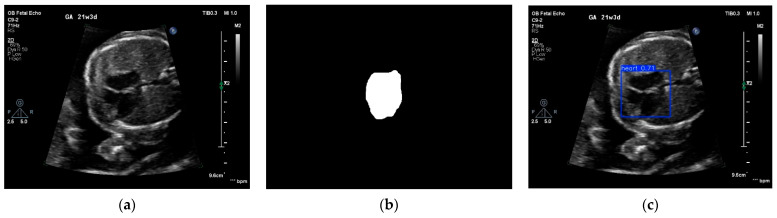
(**a**) A frame of a four-chamber view, (**b**) a segmented fetal heart by ZoomNet, and (**c**) a detected fetal heart using YOLO (version 7).

**Table 1 medicina-61-00561-t001:** A concise summary of all selected studies reviewed in Section 3. GAN is known as a generative adversarial network. Non-technical explanations of GAN and CNN can be found in Appendix A.

Titles	Input Data	Types of AI Algorithms	Application	Clinical Relevance
A Pseudo-Siamese Feature Fusion Generative Adversarial Network for Synthesizing High-Quality Fetal Four-Chamber Views [16]	Four-chamber view ultrasound images	GAN: Pseudo-Siamese Feature Fusion Generative Adversarial Network (PSFFGAN)	Synthesize echocardiograms	Enrich training database for designing better diagnostic CHD AI tools
Diagnosis of fetal total anomalous pulmonary venous connection based on the post-left atrium space ratio using artificial intelligence [17]	Four-chamber view time-resolved ultrasound videos	CNN: DeepLabv3+, FastFCN, PSPNet, and DenseASPP	Segmentation, biometric measurement, and diagnosis of specific abnormality	Diagnosis of fetal total anomalous pulmonary venous connection
Deep learning-based differentiation of ventricular septal defect from tetralogy of Fallot in fetal echocardiography images [18]	ultrasound images	CNN: VGG19, ResNet50, NTS-Net, and WSDAN	Diagnosis of abnormality	Differentiation of ventricular septal defect from tetralogy of Fallot
FetalNet: Low-light fetal echocardiography enhancement and dense convolutional network classifier for improving heart defect prediction [19]	Four-chamber view ultrasound images	CNN-based LLIE model	Image enhancement	A DL-based image processing to improve AI-based CHD detection
An ensemble of neural networks provides expert-level prenatal detection of complex congenital heart disease [20]	Ultrasound images with five standard views	CNN: Ensemble Network	Detection of cardiac views and classification of normal and abnormal hearts	Differentiate complex CHD from normal hearts
A multi-task deep learning approach for real-time view classification and quality assessment of echocardiographic images [21]	Ultrasound images with multiple views	A multi-task CNN model with four components: a backbone network, a neck network, a view classification branch, and a quality assessment branch	Selection of high-quality views	Improve automation for view selection in the clinical workflow
An intelligent quantification system for fetal heart rhythm assessment: A multicenter prospective study [22]	Ultrasound Pulse Wave Doppler	R-CNN	Calculate fetal cardiac time intervals	Assess fetal rhythm and function
A Generic Quality Control Framework for Fetal Ultrasound Cardiac Four-Chamber Planes [23]	Ultrasound images with multiple views	Varying CNN models	Quality assessment of ultrasound images	Improve automation for image selection in the clinical workflow
A progressive growing generative adversarial network composed of enhanced style-consistent modulation for fetal ultrasound four-chamber view editing synthesis [24]	Four-chamber view ultrasound images	An enhanced Generative Adversarial Network (GAN)	Synthesize echocardiograms	Enrich training database for designing better diagnostic CHD AI tools
Application of Machine Learning in screening for congenital heart diseases using fetal echocardiography [25]	Manually measured parameters	Random Forest	Detecting CHD	Screen for CHDs
Multiview and multiclass image segmentation using deep learning in fetal echocardiography [26]	Ultrasound images with multiple views	V-Net [27]	Heart structure segmentation	Improve the automation for detecting CHDs
A deep learning framework for identifying and segmenting three vessels in fetal heart ultrasound images [28]	Ultrasound images with three-vessel view	Varying U-Net [29] models	Segmenting heart structure	Improve the automation for detecting CHDs
Image segmentation of the ventricular septum in fetal cardiac ultrasound videos based on deep learning using time-series information [30]	Four-chamber view ultrasound images	CNN: Cropping–Segmentation–Calibration (CSC)	Detection of ventricular septum	Foundation for detecting VSD
Prenatal diagnosis of hypoplastic left heart syndrome on ultrasound using artificial intelligence: How does performance compare to a current screening programme [31]	Four-chamber view ultrasound images	ResNet	Detecting hypoplastic left heart syndrome	Detect hypoplastic left heart syndrome
Classification of normal and abnormal fetal heart ultrasound images and identification of ventricular septal defects based on deep learning [32]	Ultrasound images with five standard views	YOLO (version 5) and other CNN models	Classification of normal and abnormal fetal hearts	Screen fetal CHDs
Application of artificial intelligence in VSD prenatal diagnosis from fetal heart ultrasound images [33]	Ultrasound images with the four-chamber view and left ventricular outflow view	ResNet-18, DenseNet, and MobileNet,	Detect VSD	Auto-detection of VSD
Application of Artificial Intelligence in Anatomical Structure Recognition of Standard Section of Fetal Heart [34]	5 views of ultrasound images	U-Y-Net derived from YOLO (version 5)	View recognition of standard ultrasound views	Foundation for detecting CHD
Fetal Heart Disease Detection Via Deep Reg Network Based on Ultrasound Images [35]	Not specified	AlexNet, ResNet-50 VGG-16, DenseNet, MobileNet, and RegNet	Characterize normal vs. abnormal fetal hearts	Screen fetal CHDs
Prenatal Diagnosis and Fetopsy Validation of Complete Atrioventricular Septal Defects Using the Fetal Intelligent Navigation Echocardiography Method [36]	Four-chamber view ultrasound images	Spatial image correlation	Ultrasound View Navigation	Improve visualization of ventricular walls
Using Innovative Machine Learning Methods to Screen and Identify Predictors of Congenital Heart Diseases [37]	Self-reported questionnaires and routine clinical laboratory test results	Explainable Boosting Machine	Identify predictors of CHDs	Improve the detection of fetal CHDs

**Table 2 medicina-61-00561-t002:** A summary of the characteristics used in eighteen (18) papers. ASE stands for the American Society of Echocardiography. N/P and N/A denote not provided and not applicable, respectively.

Paper	Center	Vendor	Data Acquisition Protocol	Training Data Size	ASE Category [41]
Qiao et al. [16]	1	N/P	N/P	~1000	N/A
Wang et al. [17]	1	1	Minimal	~300	Assisted
Yu et al. [18]	1	N/P	N/P	~200	Autonomous
Sutarno et al. [19]	1	1	N/P	~500	Assisted
Arnaout et al. [20]	2	4	Yes	~100,000	Autonomous
Li, et al. [21]	1	4	N/P	~100,000	Autonomous
Yang et al. [22] ^1^	14	2	Yes	~10,000	Autonomous
Dong et al. [23]	1	N/P	N/P	~7000	Assisted
Qiao et al. [24]	1	N/P	N/P	~600	N/A
Truong et al. [25]	1	1	N/P	~4000	Autonomous
Wong et al. [26]	N/P	N/P	N/P	~300	Assisted
Yan et al. [28]	1	1	Minimal	~500	Assisted
Dozen et al. [30]	1	1	Minimal	~600	Assisted
Day et al. [31]	1	1	N/P	~10,000	Autonomous
Yang et al. [32]	1	3	Minimal	~1800	Autonomous
Li, et al. [33]	1	5	N/P	~1500	Autonomous
Wu, et al. [34]	1	5	N/P	~3400	Assisted
Magesh, et al. [35]	N/P	N/P	N/P	~400	Autonomous

^1^ Doppler spectra were used to detect irregular heart rhythm.

**Table 3 medicina-61-00561-t003:** A summary of the performance of four AI studies, which differentiate abnormal fetal hearts from normal fetal hearts. AUC, PPV, and NPV stand for the area under the curve, positive predictive value, and negative predictive value, respectively. N/P denotes not provided.

Paper	Accuracy	AUC	Sensitivity	Specificity	PPV	NPV
Arnaout et al. [20]	~100%	0.99	95%	96%	20%	100%
Truong et al. [25]	88%	0.94	85%	88%	55%	97%
Yang et al. [32]	80%	N/P	90%	N/P	90%	N/P
Magesh, et al. [35]	97%	0.97	92%	95%	N/P	N/P
Meta-analysis of clinical studies [3]	N/P	0.99	68.5%	99.8%	N/P	N/P

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
