# Peer review of "The Artificial Intelligence-Enhanced Echocardiographic Detection of Congenital Heart Defects in the Fetus: A Mini-Review"

_medicina, 2025, doi:10.3390/medicina61040561_

Round 1

Reviewer 1 Report

Comments and Suggestions for Authors

The manuscript addresses a highly relevant topic in medical imaging and AI, focusing on improving the detection of congenital heart defects (CHDs). It provides a comprehensive overview of the technical advancements and clinical applications of AI-enhanced echocardiography.

  1. Abstract:

    • Add one or two sentences about the ethical and practical challenges of deploying AI in low-resource settings.
  2. Introduction:

    • The introduction provides a solid background on CHDs and the need for early detection. However, it could better frame the importance of AI in addressing gaps in current diagnostic workflows.
  3. Technical Background:

    •  
    • Suggestion: Provide a brief explanation or glossary of key terms such as CNNs, GANs, and reinforcement learning for readers from a non-technical background.
  4. Applications of AI:

    • The manuscript covers a range of AI applications but lacks sufficient linkage to their practical impact on clinical outcomes.
    • Suggestion: For each application, include examples or case studies showing how AI improved diagnostic accuracy or efficiency in clinical settings.
  5. Limitations:

    • The limitations section is underdeveloped and could include more discussion on ethical, practical, and technical challenges.
    • Suggestion: Expand this section to address concerns about data bias, model generalizability, and the risk of over-reliance on AI over human expertise.
  6. Figures and Tables:

    • Figures and tables enhance the manuscript but could be labeled more clearly to improve standalone comprehension.
    • Suggestion: For example, in Table 1, add a column summarizing the clinical relevance of each AI model discussed.
  7. Language and Readability:

    • Some sentences are verbose or awkwardly phrased.
    • Suggestion: Revise for clarity and conciseness. For example, "These techniques have varying benefits, especially with image analysis and potential differences in resource draw and needs" could be simplified.
  8. References:

    • The references are extensive but could be updated to include more recent studies published in 2023–2024.
    •  
  9. Future Directions:

    • The manuscript lacks a forward-looking perspective.
    • Suggestion: Add a dedicated section on future research directions, such as integrating AI with wearable devices or telemedicine for CHD detection.
    •  
  10. Formatting:

    • Some sections, particularly the technical background, contain lengthy paragraphs that could be broken up for better readability.
    • Suggestion: Use subheadings or bullet points to organize content more effectively.
Comments on the Quality of English Language

Sentence Structure:

  • Many sentences are overly complex or verbose, making them difficult to follow.
  • Example:
    • Current: "These techniques have varying benefits, especially with image analysis and potential differences in resource draw and needs."
    • Suggested: "These techniques offer various benefits for image analysis and resource optimization."
    • Grammar and Punctuation:

      • Issues with subject-verb agreement, misplaced modifiers, and missing articles ("a," "the") are present.
      • Example:
        • Current: "Echocardiography alone has a low sensitivity for detecting CHDs, particularly when the operators’ experience is limited."
        • Suggested: "Echocardiography alone demonstrates low sensitivity in detecting CHDs, especially when operator experience is limited."
    • Clarity:

      • Some sentences lack clarity due to vague language or poor organization.
      • Example:
        • Current: "The choice of the actual ML algorithms depends on the specific clinical application and clinical data, as reviewed below."
        • Suggested: "The selection of machine learning algorithms depends on the specific clinical application and available data."
    • Redundancy:

      • Repetition of ideas in different sections reduces the overall conciseness.
      • Example: Definitions of AI methodologies like CNN and GAN are repeated multiple times.
    • Flow and Cohesion:

      • Transitions between paragraphs and sections could be smoother to improve the logical flow.
      • Example: Provide linking sentences at the beginning or end of sections to connect them with the overarching theme.

Author Response

Our detailed responses can be found in the attached  file. 

Reviewer 2 Report

Comments and Suggestions for Authors

The review is of a great interest, but as a reviewer I have to comment some issues.

General comments:

-please review profoundly the subject and perform "full" review, not mini-review

-please compare AI to human fetal echocardiography results (maybe as a Table?)

-please consider future perspectives. Would AI substitute sonographers? 

Some other primary comments are attached in the file

Comments on the Quality of English Language

It is well written according to my experience. 

Author Response

Please see our replies in the attached file below. Thank you. 

Reviewer 3 Report

Comments and Suggestions for Authors

Artificial Intelligence Enhanced Echocardiographic Detection of Congenital Heart Defects: A Mini Review

General Assessment:

This manuscript offers a comprehensive review of the application of artificial intelligence (AI) in echocardiography for detecting congenital heart defects (CHD). The subject is highly relevant, given the growing role of AI in medical imaging, and addresses a gap in leveraging advanced technologies for improving prenatal and postnatal diagnostics. However, several areas need refinement for greater clarity, scientific rigor, and alignment with publication standards.

Strengths:

  1. Relevance of Topic:
    • The manuscript covers an important and timely topic, as AI-driven tools are becoming increasingly significant in medical diagnostics, especially in underserved or resource-limited regions.
  2. Comprehensive Literature Coverage:
    • The manuscript references several recent studies, providing an up-to-date overview of AI techniques, algorithms, and their applications in echocardiography.
  3. Organization:
    • The structure is logical and clear, with distinct sections discussing technical AI background, clinical applications, and challenges.

Areas for Improvement:

1. Abstract:

  • Lack of Specificity: While the abstract broadly outlines the objectives and applications, it lacks specific quantitative data or key findings from referenced studies.
    • Example: Include metrics or examples to support statements such as "enabling untrained professionals" or "greatly benefit low-resource medical facilities."
  • Recommendation: Add concise details about AI model performance metrics and a summary of the main conclusions to make the abstract more informative.

2. Introduction:

  • Clarity of Problem Statement: The introduction briefly discusses the prevalence and significance of CHDs but lacks a focused problem statement on the limitations of current diagnostic tools.
    • Recommendation: Explicitly outline the diagnostic gaps in echocardiography that AI addresses.
  • Redundancy: Some phrases (e.g., "swift prenatal diagnosis reduces morbidity and mortality") are repeated, leading to unnecessary wordiness.

3. Methodology:

  • While this is a review paper, it would benefit from a more systematic explanation of how the literature was selected (e.g., inclusion/exclusion criteria, search terms, databases used).
    • Recommendation: Add a section detailing the review methodology to enhance transparency and reproducibility.

4. Technical Content:

  • Simplification Needed: The descriptions of AI methodologies (e.g., supervised vs. unsupervised learning, CNN architectures) are detailed but might overwhelm readers unfamiliar with AI. Simplify technical jargon or include diagrams for clarity.
  • Missing Comparisons: The paper does not adequately compare AI models in terms of performance metrics, such as sensitivity, specificity, and AUC, from the studies it cites.
    • Recommendation: Provide a table summarizing AI models, datasets, and their performance metrics for CHD detection.

5. Discussion:

  • Overgeneralization: The paper occasionally overstates AI's current capabilities (e.g., suggesting AI tools can reliably replace trained professionals). A more balanced discussion of AI's limitations and real-world applicability is needed.
  • Recommendation: Highlight potential biases, dataset limitations, and the lack of generalizability in current AI studies.

6. Language Quality:

  • Strengths: The language is generally clear and free from major grammatical errors.
  • Issues:
    • Awkward phrasing in sentences like "AI-assisted technologies have made and will continue to make an impact."
    • Overuse of technical terms without adequate context, which might confuse readers.
  • Recommendation: A professional language editor should polish the manuscript for fluency and readability.

Author Response

Please see the attached document for our detailed response. Thank you very much for the constructive comments. 

Round 2

Reviewer 1 Report

Comments and Suggestions for Authors

Weaknesses and Areas for Improvement:

  1. Limited Discussion on Clinical Integration Challenges

    • While the manuscript outlines the potential of AI in echocardiographic CHD detection, it does not discuss in-depth the challenges of integrating these models into clinical workflows, such as:
      • Regulatory approvals (e.g., FDA, CE marking)
      • Interoperability with existing hospital systems
      • Acceptance and usability by clinicians

    Recommendation:
    Expand the discussion on implementation barriers and possible solutions.

  2. Insufficient Quantitative Comparison with Human Performance

    • Although some AI models show superior sensitivity compared to human experts, there is a lack of detailed statistical comparisons in real-world clinical settings.

    Recommendation:
    Provide additional insights into studies that compare AI models directly with human echocardiographers.

  3. Unclear Data Source and Methodology in Some Sections

    • The manuscript mentions the collection of literature based on Web of Science and PubMed but lacks clear inclusion/exclusion criteria for selected studies.

    Recommendation:
    Specify the exact search strategy, keywords, and inclusion/exclusion criteria to ensure transparency and reproducibility.

  4. AI Model Generalizability Issues Not Fully Addressed

    • The manuscript highlights the importance of training AI models on diverse datasets but does not provide concrete solutions for overcoming model bias and generalizability challenges.

    Recommendation:
    Discuss potential strategies such as federated learning, data augmentation techniques, and multi-institutional collaborations to improve AI model robustness.

  5. Lack of Real-World Case Studies or Clinical Trials

    • While theoretical discussions are robust, the manuscript does not provide examples of AI models currently deployed in clinical settings.

    Recommendation:
    Include real-world case studies or discuss ongoing clinical trials validating AI-based CHD detection models.

  6. Typographical and Formatting Issues

    • Some sections have redundant explanations (e.g., convolutional neural networks are described multiple times).
    • The citation format appears inconsistent in a few places.

    Recommendation:

    • Streamline the manuscript to avoid redundancy.
    • Ensure consistency in referencing style.
Comments on the Quality of English Language

Typographical and Formatting Issues

    • Some sections have redundant explanations (e.g., convolutional neural networks are described multiple times).
    • The citation format appears inconsistent in a few places.

Author Response

Please see the attached file. Thank you for your comments, which have significantly elevated our review paper. They are highly appreciated. 

Reviewer 2 Report

Comments and Suggestions for Authors

The Authors have adjusted the text and included all the comments.

Author Response

Thank you for your kind words in the two rounds of review. We appreciate your input, which has significantly improved our work.